# DEEP TRANSFORMER Q-NETWORKS FOR PARTIALLY OBSERVABLE REINFORCEMENT LEARNING

## ABSTRACT

Real-world reinforcement learning tasks often involve some form of partial observability where the observations only give a partial or noisy view of the true state of the world. Such tasks typically require some form of memory, where the agent has access to multiple past observations, in order to perform well. One popular way to incorporate memory is by using a recurrent neural network to access the agent's history. However, recurrent neural networks in reinforcement learning are often fragile and difficult to train and sometimes fail completely as a result. In this work, we propose Deep Transformer Q-Networks (DTQN), a novel architecture utilizing transformers and self-attention to encode an agent's history. DTQN is designed modularly, and we compare results against several modifications to our base model. Our experiments demonstrate that our approach can solve partially observable tasks faster and more stably than previous recurrent approaches.

## 1 INTRODUCTION

In recent years, deep neural networks have become the computational backbone of reinforcement learning, achieving strong performance across a wide array of difficult tasks including games (Mnih et al., 2015; Silver et al., 2016) and robotics (Levine et al., 2018; Gao et al., 2020). In particular, Deep Q-Networks (DQN) (Mnih et al., 2015) revolutionized the field of deep RL by achieving super-human performance on Atari 2600 games in the Atari Learning Environment (Bellemare et al., 2013). Since then, several advancements have been proposed to improve DQN (Hessel et al., 2018), and deep RL has been shown to excel in continuous control tasks as well (Haarnoja et al., 2018; Fujimoto et al., 2018).

However, most Deep RL methods assume the agent is operating within a fully observable environment; that is, one in which the agent has access to the environment's full state information. But this assumption does not hold for many realistic domains due to components such as noisy sensors, occluded images, or additional unknown agents. These domains are *partially* observable, and pose a much bigger challenge for RL compared to the standard fully observable setting. Indeed, naïve methods often fail to learn in partially observable environments without additional architectural or training support (Pinto et al., 2017; Igl et al., 2018; Ma et al., 2020).

To solve partially observable domains, RL agents may need to remember (some or possibly all) previous observations (Kaelbling et al., 1998). As a result, RL methods typically add some sort of memory component, allowing them to store or refer back to recent observations in order to make more informed decisions. The current state-of-the-art approaches integrate recurrent neural networks, like LSTMs (Hochreiter & Schmidhuber, 1997) or GRUs (Cho et al., 2014), in conjunction with fully observable Deep RL architectures to process an agent's history (Ni et al., 2021). But recurrent neural networks (RNNs) can be fragile and difficult to train, often requiring complicated "warm-up" strategies to initialize its hidden state at the start of each training batch (Lample & Chaplot, 2017). Conversely, the Transformer has been shown to model sequences much better than RNNs and is ubiquitous in natural language processing (NLP) (Devlin et al., 2018) and increasingly common in computer vision (Dosovitskiy et al., 2020).

Therefore, we propose Deep Transformer Q-Network (DTQN), a novel architecture using self-attention to solve partially observable RL domains. DTQN leverages a transformer decoder architecture with learned positional encodings to represent an agent's history and accurately predict Q-values at each timestep. Rather than a standard approach that trains on a single next step for a given history,

we propose a training regime called intermediate Q-value prediction, which allows us to train DTQN on the Q-values generated for each timestep in the agent's observation history and provide more robust learning. DTQN encodes an agent's history more effectively than recurrent methods, which we show empirically across several challenging partially observable environments. We evaluate and analyze several architectural components, including: gated skip connections (Parisotto et al., 2020), positional encodings, identity map reordering (Parisotto et al., 2020), and intermediate value prediction (Al-Rfou et al., 2019). Our results provide strong evidence that our approach can successfully represent agents' histories in partially observable domains. We visualize attention weights showing DTQN learns an understanding of the domains as it works to solve tasks.

## 2 BACKGROUND

When an environment does not emit its full state to the agent, the problem can be modeled as a Partially Observable Markov Decision Process (POMDP) Kaelbling et al. (1998). A POMDP is formally described as the 6-tuple $(\mathcal{S}, \mathcal{A}, \mathcal{T}, \mathcal{R}, \Omega, \mathcal{O})$. $\mathcal{S}$, $\mathcal{A}$, and $\Omega$ represent the environment's set of states, actions, and observations, respectively. $\mathcal{T}$ is the state transition function $\mathcal{T}(s, a, s') = P(s'|s, a)$, denoting the probability of transitioning from state $s$ to state $s'$ given action $a$. $\mathcal{R}$ describes the reward function $\mathcal{R} : \mathcal{S} \times \mathcal{A} \rightarrow \mathbb{R}$; that is, the resultant scalar reward emitted by the environment for an agent that was in some state $s \in \mathcal{S}$ and took some action $a \in \mathcal{A}$. And $\mathcal{O}$ is the observation function $\mathcal{O}(s', a, o) = P(o|s', a)$, the probability of observing $o$ when action $a$ is taken resulting in state $s'$. At each time step, $t$, the agent is in the environment's state $s_t \in \mathcal{S}$, takes action $a_t \in \mathcal{A}$, manipulates the environment's state to some $s_{t+1} \in \mathcal{S}$ based on the transition probability $\mathcal{T}(s_t, a_t, s_{t+1})$ and receives a reward, $r_t = \mathcal{R}(s_t, a_t)$. The goal of the agent is to maximize $\mathbb{E}\left[\sum_t \gamma^t r_t\right]$, its expected discounted return for some discount factor $\gamma \in [0, 1)$ (Sutton & Barto, 2018).

Because agents in POMDPs do not have access to the environment's full state information, they must rely on the observations $o_t \in \Omega$ which relate to the state via the observation function, $\mathcal{O}(s_{t+1}, a_t, o_t) = P(o_t|s_{t+1}, a_t)$. In general, agents acting in partially observable space cannot simply use observations as a proxy for state, since several states may be aliased into the same observation. Instead, they often consider some form of their full history of information, $h_t = \{(o_0, a_0), (o_1, a_1), ..., (o_{t-1}, a_{t-1})\}$. Because the history grows indefinitely as the agent proceeds in a trajectory, various ways of encoding the history exist. Previous work has truncated the history to make it a fixed length (Zhu et al., 2017) or used an agent's belief, which represents the estimate of the current state (Kaelbling et al., 1998). Since the deep learning revolution, others have used forms of recurrency, such as LSTMs and GRUs, to encode the history (Hausknecht & Stone, 2015; Yang & Nguyen, 2021).

### 2.1 DEEP RECURRENT Q-NETWORKS

Q-Learning (Watkins & Dayan, 1992) aims to learn a function $Q : \mathcal{S} \times \mathcal{A} \rightarrow \mathbb{R}$ which represents the value of each state-action pair in an MDP. Given a state $s$, action $a$, reward $r$, next state $s'$, and learning rate $\alpha$, the $Q$-function is updated with the equation

$$Q(s, a) := Q(s, a) + \alpha(r + \max_{a' \in \mathcal{A}} Q(s', a') - Q(s, a)) \tag{1}$$

In more challenging domains, however, the state-action space of the environment is often too large to be able to learn an exact $Q$-value for each state-action pair. Instead of learning a tabular Q-function, DQN (Mnih et al., 2015) learns an approximate $Q$-function featuring strong generalization capabilities over similar states and actions. DQN is trained to minimize the Mean Squared Bellman Error

$$L(\theta) = \mathbb{E}_{(s,a,r,s') \sim \mathcal{D}}\left[\left(r + \max_{a' \in \mathcal{A}} Q(s', a'; \theta') - Q(s, a; \theta)\right)^2\right] \tag{2}$$

where transition tuples of states, actions, rewards, and future states $(s, a, r, s')$ are sampled uniformly from a replay buffer, $D$, of past experiences while training. The target $r + \max_{a' \in \mathcal{A}} Q(s', a'; \theta')$ invokes DQN's target network (parameterized by $\theta'$), which lags behind the main network (parameterized by $\theta$) to produce more stable updates.

However, in partially observable domains, DQN may not learn a good policy by simply replacing the network's input from states to observations (i.e., an agent can often perform better by remembering

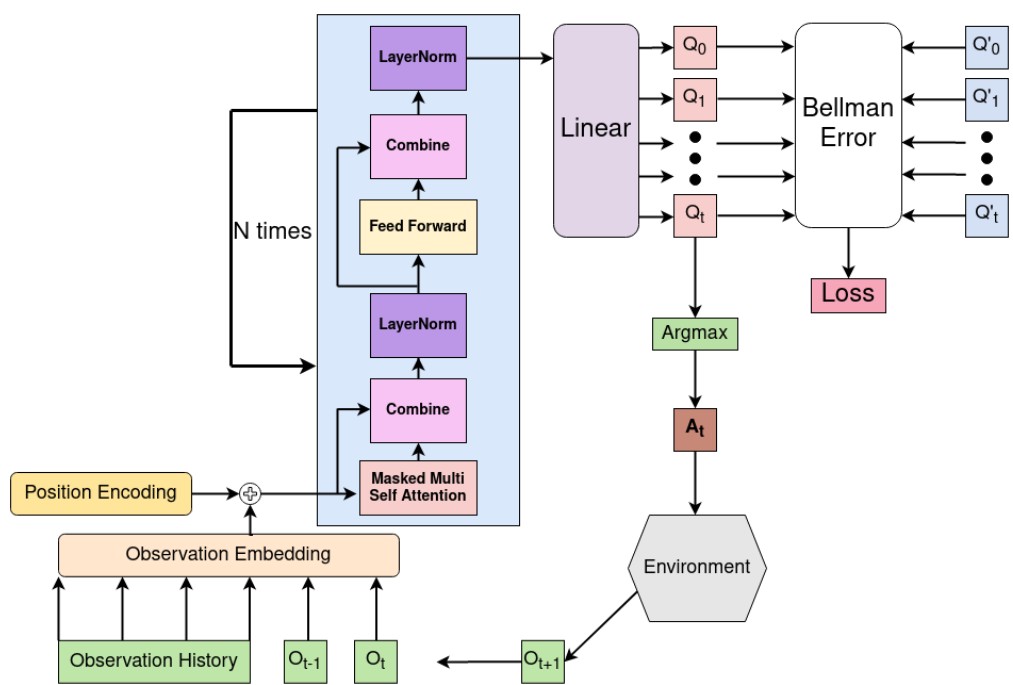

Figure 1: Architectural diagram of DTQN. Each observation in the history is embedded independently, and Q-values are generated for each observation sub-history. Only the last set of Q-values are used to select the next action, but the other Q-values can be utilized for training.

some history). To address this challenge, Deep Recurrent Q-Networks (DRQN) (Hausknecht & Stone, 2015) incorporated histories into the $Q$-function by way of a long short-term memory (LSTM) layer (Hochreiter & Schmidhuber, 1997). In DRQN's training procedure, the sampled states are replaced with histories $h_{t:t+k} = \{o_t, o_{t+1}, ..., o_{t+k}\}$ from timestep $t$ to step $t + k$, sampled randomly within each episode. The hidden state of the LSTM is zeroed at the start of each update.

## 2.2 TRANSFORMER DECODERS

The transformer architecture (Vaswani et al., 2017), originally introduced for natural language processing, stacks blocks of attention layers (Bahdanau et al., 2014) and is typically used to model sequential data. Intuitively, the transformer's attention module receives as input a sequence of tokens (e.g., a sequence of observations in an episode) and the model learns to place stronger weights or more *attention* on the most important tokens. For more details about the attention module in transformers, refer to Appendix A.3.

While the original transformer architecture formed an encoder-decoder structure, recent works often use either the encoder (Devlin et al., 2018) or the decoder (Radford et al., 2018). The key difference between the two is that the decoder applies a causal masking to the attention layer; that is, the $ith$ token cannot attend to tokens which come later in the sequence. In general, the transformer decoder has been shown to perform better on generative tasks like next token prediction, while the transformer encoder is able to learn richer representations and excels on tasks such as language understanding.

DTQN utilizes the transformer decoder structure. Given a tensor of shape $(B, C, D)$, where $B$ is the batch size, $C$ is the context length, and $D$ is the model's dimensionality size, the transformer decoder layer returns a tensor of the same shape, enabling us to stack layers on top of each other. The last transformer layer's output can then be projected to the desired shape, or sent as input to another network. To ensure the raw inputs are of the correct shape, we often prepend a feature extraction step, such as a lookup embedding for text or integers, a multilayer perceptron for vectors, or convolutional neural network for images.

## 3 RELATED WORK

Since its inception, several works have built upon DRQN. For example, DRQN was shown to beat human test subjects on the challenging 3D VizDoom video game environment (Kempka et al., 2016) when augmented with game feature information (Lample & Chaplot, 2017). *Action-based* DRQN (ADRQN) (Zhu et al., 2017) conditioned the network on the agent's full history rather than just its observation history. Deep Distributed Recurrent Q-Networks (DDRQN) (Foerster et al., 2016) extends DRQN into the multi-agent reinforcement learning setting. Like ADRQN, DDRQN conditioned on actions, but also shared weights between each agent, all while forgoing each agent's replay buffer to sample from.

The concept of attention is also well-studied in the reinforcement learning setting. The most closely related work to ours using attention in deep Q-learning is Deep Attention Recurrent Q-Network (DARQN) (Sorokin et al., 2015), which used attention to aid an LSTM's representation of the agent's history. Similarly, visual attention has been added to DRQN-like architectures in an effort towards creating more interpretable reinforcement learning algorithms (Mott et al., 2019). Unlike our work, which uses self-attention such that agent's history forms the queries, keys, and values, these works use the recurrent network's last output state to form the queries, and the environment's most recent observation forms the keys and values. In the multi-agent setting, Multi-Actor-Attention-Critic (Iqbal & Sha, 2019) created an attention module in which each agent's query is their own observation, and the keys and values are formed by the other agents' observations. Finally, the Simple Neural AttentIve Learner (SNAIL) (Mishra et al., 2017) used attention to develop a meta-learning agent capable of transferring its skills to similar but different environments.

The use of transformers in reinforcement learning has become more popular within the last few years. In the offline reinforcement learning setting, Decision Transformer (Chen et al., 2021) and Trajectory Transformer (Janner et al., 2021) concurrently proposed the idea of using transformer decoders for sequence modeling, surpassing the current offline RL state of the art. Online Decision Transformer (Zheng et al., 2022) extended Decision Transformer (Chen et al., 2021) by treating the offline training as a pre-training step, and fine-tuned the transformer in the online setting for even better performance. FlexiBiT (Carroll et al., 2022) trained a transformer encoder to learn a variety of inference tasks, such as behavioural cloning, forward and backward modeling, and inferring an agent's history given its current state. Contrary to our work, which is trained completely online using reinforcement learning, these works are specialized to take advantage of an offline RL dataset, and train their agents in a supervised way (Schmidhuber, 2019). Other methods use transformers to learn from scratch in the online RL setting, like GTrXL (Parisotto et al., 2020), which modifies the ordering of components within the transformer block, and introduces a new gating mechanism to replace the residual skip connections. We compare the effects of these modifications to our architecture. Lightweight transformers have shown strong performance in text adventure games (Xu et al., 2020), and the transformer encoder was applied to video games (Upadhyay et al., 2019). In contrast to these works, we utilize a multi-layer transformer decoder architecture. Vision transformers have been used in conjuction with DQN to stabilize Q-learning with data augmentation, replacing the standard convolutional neural networks (Hansen et al., 2021). The self-attention block in their work attends to features within a single observation whereas ours attends throughout the agent's history.

## 4 DEEP TRANSFORMER Q-NETWORK ARCHITECTURE

Transformers seem like a natural fit to represent histories in POMDPs, but there are several open questions regarding how to use them best in deep RL. In particular, it is unclear what form of transformer to use, how to integrate it into deep RL methods and how they should be trained. We chose to build DTQN using a transformer decoder structure incorporating learned position encodings, and train on the Q-values generated for each timestep in the agent's observation history. DTQN takes as input the agent's previous $k$ observations, $h_{t:t+k} = \{o_t, o_{t+1}, ..., o_{t+k-1}\}$, linearly projects each observation into the dimensionality of the model, and adds positional encodings to add information about the absolute temporal location of each observation. The embedded history is then passed through $N$ transformer layers, and finally projected to the action space of the environment (see Figure 1 and Algorithm 1). DTQN outputs a set of Q-values relating to each observation in the input.

---

**Algorithm 1** DTQN

---

**function** FORWARD PASS($h_{t:t+k} = \{o_t, o_{t+1}, ..., o_{t+k-1}\}$))
    $E^0 = \text{Embedding}(h_{t:t+k}) + Pos$
    **for** Layer $L = 1, ..., N$ **do**
        $Q^{L-1} = E^{L-1}W_{L-1}^Q, K^{L-1} = E^{L-1}W_{L-1}^K, V^{L-1} = E^{L-1}W_{L-1}^V$
        $E^L = \text{LayerNorm}_1^L(\text{Combine}_1^L(\text{MultiHeadAttention}^L(Q^{L-1}, K^{L-1}, V^{L-1}), E^{L-1}))$
        $E^L \leftarrow \text{LayerNorm}_2^L(\text{Combine}_2^L(\text{FFN}^L(E^L), E^L))$
    **end for**
    $Output \leftarrow \text{FFN}^N(E^N)$                              ▷ Project output to action space
**end function**
**function** TRAIN
    Sample a minibatch of contexts $(h_{t:t+k}, a_{t:t+k}, r_{t:t+k}, h_{t+1:t+k+1})$ from replay buffer $D$
    **for** $i = 1, ..., k$ **do**
        $L_i(\theta) = \mathbb{E}_{(.)\sim D}\left[\left(r_{t+i-1} + \max_{a'\in\mathcal{A}} Q(h_{t+1:t+i+1}, a'; \theta') - Q(h_{t:t+i}, a_{t+i-1}; \theta)\right)^2\right]$
    **end for**
**end function**
**function** UPDATE
    $\theta \leftarrow \theta - \alpha\nabla_\theta \sum_{i=1}^k L_i(\theta)$
**end function**

---

While we only use the Q-values from the most recent observation during execution, we train the network using all generated Q-values, even those relating to the observations at the beginning of the subhistory using the loss function in Algorithm 1. This training regime challenges the network to predict the Q-values in situations where it has little to no context, and produces a more robust agent. The remainder of this section expands on each contribution of the DTQN architecture.

## 4.1 OBSERVATION EMBEDDINGS AND POSITIONAL ENCODINGS

Before the observation history is passed to DTQN's transformer layers, each observation in the agent's most recent $k$ observations, $h_{t:t+k}$, is linearly projected to the dimensionality of the transformer via a learned observation embedding (see Figure 1). After embedding, we add a learned positional encoding to each observation based on its position in the observation history. This result, which we call $E^0$ in Algorithm 1, is the input to the first transformer layer in DTQN.

Position encodings are common practice in transformers, especially for NLP tasks, where they are well studied (Wang & Chen, 2020). However, the importance of position is less clear in the reinforcement learning setting. In some control tasks, the temporal position of an observation may not have any effect on its importance or meaning to solve the task. For instance, the importance of the priest observation in the classic HeavenHell domain (Bonet, 1998) is not dependent on when the observation occurs in the episode. On the other hand, domains with more dynamic state transitions may benefit greatly from the positional information. For this reason, we choose to learn our positional encodings as it gives the agent the most flexibility in terms of how it chooses to use them. We ablate this choice by comparing our learned positional encodings to sinusoidal positional encodings (used in the original transformer (Vaswani et al., 2017)) as well as not using any positional encodings in section 5.4.

## 4.2 TRANSFORMER DECODER STRUCTURE

Like the original GPT architecture (Radford et al., 2018), each transformer layer in DTQN features two submodules: masked multi-headed self-attention and a position-wise feedforward network. As described in Algorithm 1, first we project the output of the previous layer, $E^{L-1}$ to the queries, $Q$, keys, $K$, and values, $V$, through the weight matrices $W^Q$, $W^K$, and $W^V$, respectively. After each submodule, that submodule's input and output are combined (see the "Combine" step in Figure 1) followed by a LayerNorm (Ba et al., 2016). Finally, after the last transformer layer, we project the final embedding ($E^N$ in Algorithm 1) to the action space of our environment to represent the Q-value for each action.

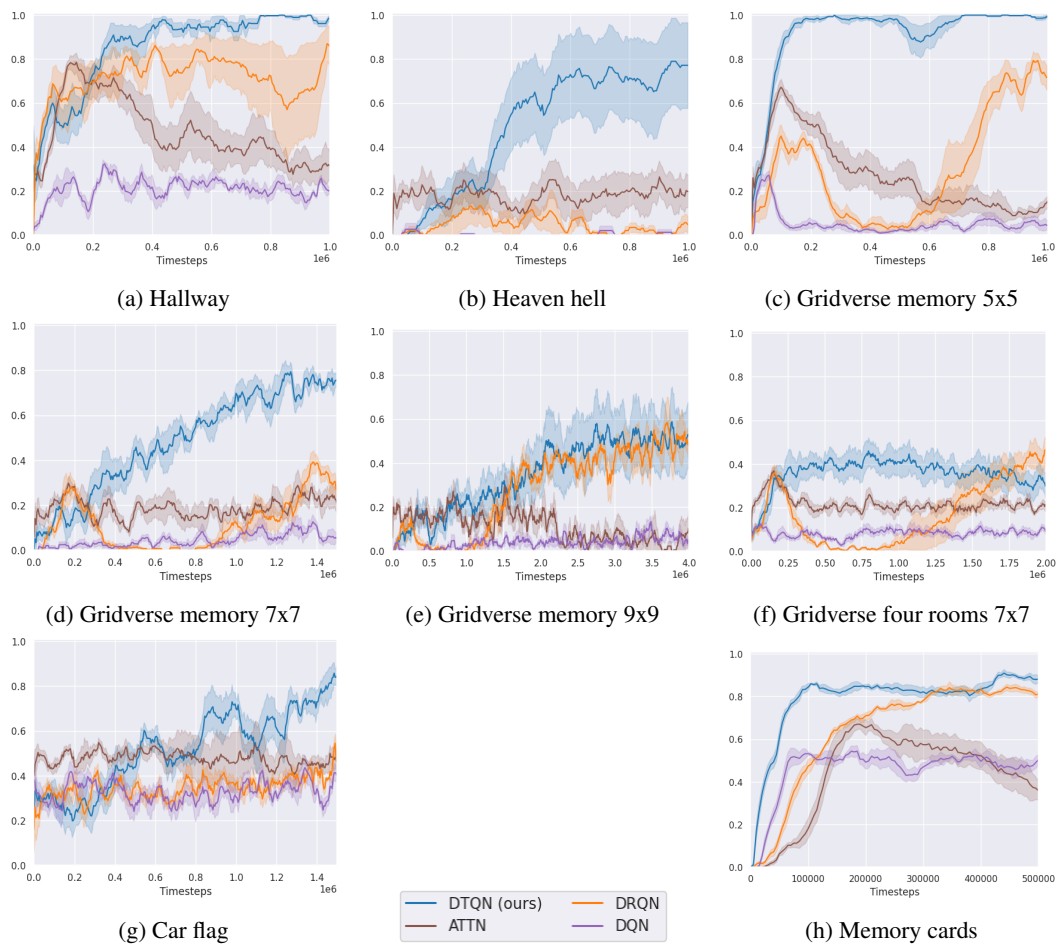

Figure 2: Results showing the success rate of DTQN against baselines. DTQN is shown in blue, a simple attention network (ATTN) shown in brown, Deep Recurrent Q-Network (DRQN) (Hausknecht & Stone, 2015) is shown in orange, and Deep Q-Network (DQN) (Mnih et al., 2015) is shown in purple. Lines show the mean and shaded regions represent standard error across 5 random seeds. DTQN excels both in terms of learning speed as well as final performance, clearly outperforming the baselines on nearly all domains. Refer to section 5.2 for discussion of results.

DTQN uses a residual skip connection (He et al., 2016) to combine the two streams, matching the original transformer, in favor of other choices of combination layers such as the GRU gating combination layer Parisotto et al. (2020). Another contested decision is the position of LayerNorm with respect to each submodule; the original transformer (Vaswani et al., 2017) and original GPT (Radford et al., 2018) apply LayerNorm after the combine step whereas other works have moved the LayerNorm to immediately before the submodule (Radford et al., 2019; Parisotto et al., 2020; Xu et al., 2020). DTQN applies the LayerNorm after the combine step, a choice we found to be simple while also demonstrating strong empirical performance. We ablate our choices of network with the aforementioned variants in section 5.3.

## 4.3 INTERMEDIATE Q-VALUE PREDICTION

DTQN outputs a set of Q-values for each timestep in the agent's observation history. During evaluation, DTQN selects the action with the highest Q-value from the last timestep in its history. It would therefore be straightforward to train DTQN using just the last timestep's Q-values, since those have the most context to work with and are the most informed to select the optimal action. This regime, however, is very wasteful, as only a fraction of the generated Q-values actually get used for training. Instead, we train DTQN using all generated Q-values. Originally used in the NLP

setting where each position was tasked with predicting the next character and formed an auxiliary loss (Al-Rfou et al., 2019), we adapt this training regime to the reinforcement learning setting, as shown in Algorithm 1. Note that the for loop depicted in Algorithm 1 can be done in one forward pass of the network because of the causally-masked self-attention mechanism.

We ablate training based on all Q-values with training only on the last timestep's Q-values in section 5.5, and show the performance gains in Table 1.

## 5 EXPERIMENTS

Our experimental evaluation is designed to compare DTQN not only to previous Q-network baselines, but also to ablate our own method with other architectural choices. We evaluate these methods on a range of challenging domains featuring partial observations and requiring memory to solve them. We baseline DTQN against Deep Recurrent Q-Networks (DRQN) (Hausknecht & Stone, 2015) to show the transformer is a more effective history representation module than RNNs, Deep Q-Networks (DQN) (Mnih et al., 2015) to demonstrate the need for memory to solve the task consistently, and against an attention baseline (called "ATTN" in Figure 2) to show the benefits of our architectural choices. ATTN, like the transformer, has observation and position embeddings, attention and feedforward network modules, but does not have LayerNorm or skip connections, and does not stack multiple blocks.

### 5.1 DOMAINS

We conduct our experiments on 4 different environment sets designed to challenge DTQN in different ways: classic POMDPs, gym-gridverse (GV) (Baisero & Katt, 2021), car flag (Nguyen, 2021), and memory cards. Hallway (Littman et al., 1995) and HeavenHell (Bonet, 1998) are classic navigation POMDPs requiring the agent to take and remember several information gathering steps before it can consistently achieve its goal. Gym-Gridverse offers procedurally generated gridworlds containing difficult partially observable tasks. The agent's field of view is restricted such that it can only see the cells in a $2 \times 3$ grid in front of it (see Figure 3), which introduces state aliasing and forces the agent to gather localizing information before it can successfully solve the task. We evaluate our agents in gridverse environments "Memory" and "Memory Four Rooms", which require the agent to first find the colored information beacons, and

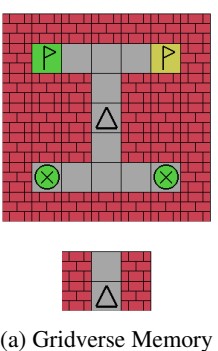 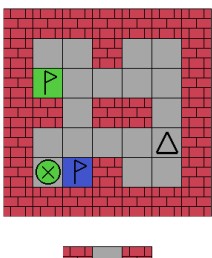

(a) Gridverse Memory 7x7

(b) Gridverse Memory Four Rooms 7x7

Figure 3: Gym-Gridverse Memory domains. The top row depicts the state while the bottom row shows the agent's current observation. The colored beacon informs the agent which flag to reach.

then go to the flag whose color matches the beacon. The colors of the flags and beacons are initialized randomly and, in Memory-Four-Rooms, the locations of the flag and beacons are also initialized randomly, increasing the environment's difficulty. Example screenshots of the gridverse domains are shown in Figure 3. Car Flag features a car on a 1D line, where the car must first drive to an oracle flag to learn which direction the finish line is. Memory cards is a novel domain designed to test how much information an agent can memorize. Based on the popular children's memory card game, 5 pairs of cards are hidden to the agent, with one card revealed at each timestep and the agent must guess the position of that card's pair. We chose this set of domains to be representative of challenging partially observable problems. For more information regarding the domains, please see Appendix B.

### 5.2 BASELINE COMPARISON

Our evaluation against baselines is shown in Figure 2. Each learning curve shows the success rate of the agent in the environment, where the line is the mean across 5 random seeds, and the shaded region represents standard error. We search across hyperparameters of interest, and select the best

performing hyperparameter set, prioritizing consistency across domains. For specific hyperparameters and training details, refer to Appendix A.

The results in Figure 2 show DTQN outperforms the baselines in terms of learning speed and final performance on nearly all domains. ATTN learns quickly, but rarely reaches optimal performance, and often becomes unstable. DRQN, featuring an LSTM as its memory module, often performs well in our set of domains, but learns slower than DTQN, and is in general less stable. Sometimes, especially in the gridverse memory domains, DRQN's performance plummets shortly after it begins to learn. It then struggles to regain its initial performance gains, taking as many as one million timesteps in the Gridverse memory 7x7 domain to improve its success rate. DQN, designed for MDPs and without any form of memory in its architecture, fails to achieve higher than 50% success rate on any of our domains, exemplifying the difficulty of the domains and the importance of using memory to solve them. These results highlight the effectiveness of DTQN in solving a range of POMDPs.

### 5.3 GRU-GATES AND IDENTITY MAP REORDERING

In this section, we compare DTQN with different forms of the "Combine" step (see Figure 1) as well as different positions of LayerNorm. DTQN's combine step is a residual skip connection, and the LayerNorm occurs after both the attention and the feedforward submodules. In contrast, GTrXL (Parisotto et al., 2020) introduced GRU-like gating in the combine step, and identity map reordering, which moves the LayerNorms directly in front of the masked multi self attention and feedforward sections. We compare our DTQN with residual skip connection to a version of DTQN which uses GRU-like gating, a version of DTQN which uses identity map reordering, and a version which uses both GRU-like gating and identity map reordering. When both GRU gates and the identity map reordering is used, the architecture resembles GTrXL. However, we do not use the TransformerXL (Dai et al., 2019) as in GTrXL, therefore we are not comparing to an exact replica.

The results for this ablation are shown in Table 1. DTQN performs competitively with the ablated versions. The variant with only identity map reordering performs significantly worse than the other versions, and the version with both identity map reordering and GRU-like gating performs worst on hallway. Both DTQN and the GRU-like gating variant perform competitively on all three domains we tested. Although we do not use the TransformerXL in our experiments, we would expect to see the same relative performance across if we replaced our transformer decoder with the TransformerXL. A comparison of different transformer backbones, such as Big Bird (Zaheer et al., 2020), sparse transformers (Child et al., 2019), or the TransformerXL would be interesting future study.

### 5.4 POSITIONAL ENCODINGS

DTQN uses learned positional encodings to allow the network to adapt to different domains. Partially observable domains will exhibit a broad range of temporal sensitivity, and we want to provide DTQN the flexibility to learn encodings to match its domain. In this section, we compare the use of learned positional encodings with the sinusoidal encodings in the original transformer (Vaswani et al., 2017) as well as no positional encodings. The results for this comparison are shown in Table 1. In the memory cards domain, the variant of DTQN without positional encodings performs significantly worse than both our learned encodings as well as the sinusoidal encodings. However, in the gridverse memory task and hallway, the three styles of positional encodings perform comparably. We analyze the resulting learned positional encodings from our trained DTQN agents across various domains in Appendix E.

### 5.5 INTERMEDIATE Q-VALUE PREDICTION

DTQN predicts and trains on the Q-values generated for each timestep in the agent's observation history. During evaluation, however, we only consider the last timestep's Q-values when determining which action to take. We could, therefore, train in the same way, only training with the last timesteps' Q-values. We compare these two training regimes, and the results for this are shown in Table 1. Our results show the variant trained without intermediate Q-values suffers a significant performance decrease. In the memory cards case, DTQN excels and solves the task with nearly 90% success rate, but the variant without intermediate Q-value prediction can barely solve the task 10% of the time. By training on all generated Q-values, we produce a more robust and effective agent.

Table 1: Ablations. We report the final success rate for each variant, averaged across 5 seeds, with standard error.

| | | GV memory 7x7 | Memory cards | Hallway | Average |
|---|---|---|---|---|---|
| Transformer structure | DTQN (ours) | $75.2 \pm 7.2$ | $89.8 \pm 1.9$ | $98.3 \pm 1.0$ | 87.77 |
| | Gate and identity | $\mathbf{80.3} \pm 6.4$ | $\mathbf{90.8} \pm 3.0$ | $67.5 \pm 10$ | 79.53 |
| | Gate only | $78.3 \pm 4.4$ | $88.9 \pm 1.2$ | $\mathbf{100} \pm 0$ | $\mathbf{89.07}$ |
| | Identity only | $65.3 \pm 6.7$ | $88.5 \pm 1.8$ | $69.8 \pm 10$ | 74.5 |
| Positional encodings | Learned (ours) | $75.2 \pm 7.2$ | $\mathbf{89.8} \pm 1.9$ | $98.3 \pm 1.0$ | $\mathbf{87.77}$ |
| | Sinusoidal | $83.1 \pm 5.0$ | $85.5 \pm 1.9$ | $92 \pm 3.9$ | 86.87 |
| | None | $\mathbf{85.5} \pm 3.7$ | $70.8 \pm 2.8$ | $\mathbf{99} \pm 0.4$ | 85.1 |
| Intermediate Q-value prediction | DTQN (ours) | $\mathbf{75.2} \pm 7.2$ | $\mathbf{89.8} \pm 1.9$ | $\mathbf{98.3} \pm 1.0$ | $\mathbf{87.77}$ |
| | None | $56.27 \pm 14$ | $9.0 \pm 2.2$ | $92.8 \pm 0.7$ | 52.69 |

## 6 DISCUSSION

DTQN outperforms or is competitive with our baselines in terms of learning speed and final performance on all our domains. One additional advantage of transformers is the ability to visualize self-attention weights as a form of interpreting the model. Intuitively, the self-attention mechanism allows the agent to prioritize observations in its history which provide it with the most information useful in solving its task. The causal masking ensures the agent cannot attend to observations in its future, resembling how the agent will need to perform during execution. While the use of attention weights as a tool for explainability is still being studied (Jain & Wallace, 2019; Wiegreffe & Pinter, 2019), it does allow us to observe which observations the agent finds most valuable in its history. In Figure 4, we visualize a trained DTQN agent's attention weights from a trajectory in gridverse memory 7x7. Crucially, the observation including the green beacon (circle with X, magnified on right) is strongly attended to by all future observations,

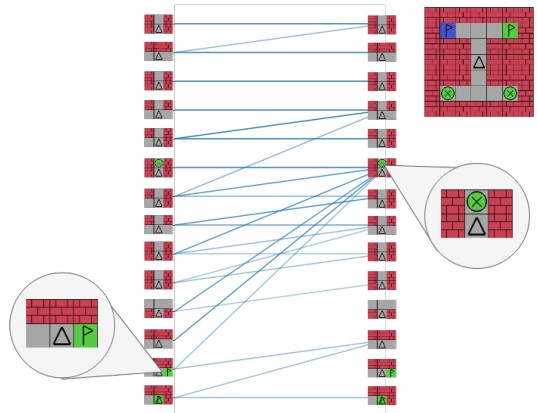

Figure 4: Attention bars for gridverse memory 7x7. Bars go from left to right, and observations go top to bottom (i.e. the second observation attended to the first and second observation). Attention weights below 0.2 have been removed for visibility.

indicating the DTQN agent has correctly learned which observations are important in solving the task. When the agent sees the green flag (magnified on left), it attends to the observation with green beacon to ensure it selects the correct flag. We provide additional attention visualizations in Appendix D

## 7 CONCLUSION

In this work we introduce Deep Transformer Q-Networks, a novel architecture for solving challenging partially observable domains with reinforcement learning. DTQN incorporates the transformer decoder, which excels in generating Q-values for each timestep of the agent's observation history. We train the model on all generated Q-values, enabling an efficient training regime and faster learning. DTQN also utilizes learned positional encodings, empowering the model to learn domain-specific encodings which match the temporal dependencies of the environment. We explore and ablate several architectural structures, and find our choices either outperform or are at least competitive with all tested variants. Finally, we provide a modular code implementation of DTQN that is easy to extend and modify, which we hope the research community will be able to use as we expect our approach to serve as the basis and benchmark for future transformer-based methods in partially observable reinforcement learning.

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
