# OpenReview forum: "Deep Transformer Q-Networks for Partially Observable Reinforcement Learning"
_ICLR.cc/2023/Conference — Submitted to ICLR 2023_

### Official Review · Reviewer_QsGx · 2022-10-17

**Confidence:** 4
**Correctness:** 4
**Technical Novelty And Significance:** 3
**Empirical Novelty And Significance:** 3
**Recommendation:** 6

**Clarity, Quality, Novelty And Reproducibility:**

The paper is clear and well written. The novelty seems to be limited to the intermediate Q-value prediction, which can still be a significant contribution, especially if there were more analysis into why it helps performance. The paper states that code is provided, but I could not see it in the supplementary materials.

**Strength And Weaknesses:**

Strengths
- The problem that the method aims to tackle, RL in partially observable environments, is an important one, and the method itself is well-motivated, i.e. leveraging the transformer architecture to predict many Q-values at once for a denser training signal.
- The approach is explained well, along with an informative diagram in Figure 1.
- The related work section is strong, explicitly placing and contrasting the method in context of other transformer-based RL methods.
- The tasks seem to be well-chosen, in the sense that they are partially observable and require some for of memory to complete.
- The choice of baselines is reasonable, particularly when considered in combination with the ablations where, for example, the “Gate and identity” ablation is highlighted to be almost the same architecture as the GTrXL (just without the memory buffer) and thus serves as a relevant comparison to the state of the art in transformer architecture for RL.

Opportunites for improvement
- Ablations / analyses. While a number of ablations were performed and described in the paper, they might be improved in the following ways:
    - In Table 1, it is shown that using gating in the combine step gives a better performance on average than the residual skip connection used in DTQN - given this, why is gating not used in the final DTQN architecture?
    - While the ablations for the combine step and the positional encodings are interesting to see, they seem tangential to the key contribution of the method, which is the intermediate Q-value predictions. Indeed an ablation is performed on the intermediate predictions, demonstrating its significant effect on performance, but there could be more investigation into *how* it helps. For example, how much of the benefit of intermediate Q-value training comes from the extra training signal that comes from training on multiple states at once, versus a potential regularising effect that comes from the Q-value predictions of earlier states in the sequence having far less context than those later in the sequence? Perhaps this could be determined by running experiments that fix the number of predicted Q-values in training, but vary how much temporal context is used for each Q-value prediction.
- While in the introduction, it is clear what the contribution of the method is (the intermediate Q-value prediction), throughout the paper the other architectural choices (which as far as I am aware are not novel, e.g. GTrXL used causal masking, positional encodings, but please correct me if I’m wrong) act as a bit of a distraction to this. I think the paper would be stronger if it focused more on the key contribution and understanding how it helps performance, as mentioned above.

Minor comments
- Table 1 it would be useful to bolden the top performing models on each task
- Section 4.3 last line: “Figure 1” -> “Table 1”
- Legend in Figure 2 quite small
- Appendix E links to higher up in the paper
- The naming of the ‘combine’ step is confusing, why not just call it a residual skip connection?

**Summary Of The Paper:**

This paper proposes a deep reinforcement learning method for partially observable environments named Deep Transformer Q-Networks (DTQN). The key feature of DTQN is that, for a given chunk of history sampled from the replay buffer, a loss is taken over predicted Q-values for *all* timesteps in the history (rather than just the last one) by applying a causal mask in the transformer and thus efficiently providing a denser training signal for the agent. The architecture is evaluated on a number of tasks in partially observable environments and compared to a number of deep Q-learning baselines (one with no memory, one RNN-based, and an attention-based), in which DTQN is shown to perform best on average. A number of ablations are also conducted, demonstrating the importance of intermediate Q-value prediction as well as architectural choices such as positional encodings and how the attention outputs are combined.

**Summary Of The Review:**

This paper proposes a method with a clear motivation for tackling the important problem of efficient RL in partially observable environments. The key contribution of intermediate Q-value prediction can be an important one (and is empirically shown to give a boost to performance in relevant tasks), but more analysis is ideally required to demonstrate how it helps performance, and the ablations, while interesting, serve as a bit of a distraction to the main method.

---

> ### Author Response · Authors · 2022-11-11
> **Response to reviewer QsGx**
>
> We thank the reviewer for their helpful comments. We hope to address the mentioned concerns below:
>
> > In Table 1, it is shown that using gating in the combine step gives a better performance on average than the residual skip connection used in DTQN - given this, why is gating not used in the final DTQN architecture?
>
> In Table 1 and Figure 6 (Appendix), we notice the gating can lead to marginally better final performance. One possible explanation for this is the gating mechanism includes more trainable parameters than the residual skip connection, giving the network more power. For simplicity and connection to the original transformer, we decided to keep the residual skip connection as the “baseline” DTQN model, but we provide an open source and modular implementation (see updated supplementary) allowing researchers to change the “combine” step (as defined in Figure 1) with a simple command line argument. Regardless, all of these variants are potentially useful versions of our approach.
>
>
> > Perhaps this could be determined by running experiments that fix the number of predicted Q-values in training, but vary how much temporal context is used for each Q-value prediction.
>
> We agree, gaining more insights into the success of intermediate Q-value prediction would be very interesting, and an potential avenue for future work.
>
> > Table 1 it would be useful to bolden the top performing models on each task
>
> We have bolded the highest-performing mean result on each of the tasks and the average in the updated version of the paper.
>
> > Section 4.3 last line: “Figure 1” -> “Table 1”
>
> Thank you for pointing this out, it has been fixed in the updated version of our paper.
>
> > Legend in Figure 2 quite small
>
> We have increased the legend size for readability in the updated version of our paper.
>
> > The naming of the ‘combine’ step is confusing, why not just call it a residual skip connection?
>
> We want to highlight the modularity of the transformer architecture. While the base DTQN implementation uses the standard residual skip connection, we also experiment with the gating mechanism from GTrXL, which we found also performs well. We generalize that area of the architecture to the “combine” step
>
> > The paper states that code is provided, but I could not see it in the supplementary materials.
>
> You can now find our source code in the updated supplementary materials.

---

> ### Author Response · Authors · 2022-11-23
> **Response to reviewer QsGx**
>
> We hope to have addressed your concerns in our response, and encourage you to continue to join in the discussion so we may address any additional questions or concerns you may have. We have listed the major updates to our paper in the top-level comment [here](https://openreview.net/forum?id=cddqs4kvC20&noteId=SHB2K4TvbF). Thank you

---

### Official Review · Reviewer_oBw9 · 2022-10-19

**Confidence:** 4
**Correctness:** 3
**Technical Novelty And Significance:** 3
**Empirical Novelty And Significance:** 2
**Recommendation:** 6

**Clarity, Quality, Novelty And Reproducibility:**

The proposed architecture is novel for solving POMDP problems and the contributions are clearly presented. The code is also provided.

**Strength And Weaknesses:**

Strength:
1. The proposed transformer-based architecture is novel and the most effective one for DQN baselines.
2. The paper is clearly presented and well-written.
3. The experimental results demonstrate the effectiveness of DTQN in solving POMDP problems.

Weakness:
1. The proposed method is only compared with DQN and DRQN while there lacks evidence to show that DTQN can lead to state-of-the-art performance, which makes it less convincing.
2. More discussions and comparison results between [1] and DTQN should be provided.
3. [2] also presents a very similar architecture (but it uses transformer encoder). More reviews and comparison results should be added.
4. GRU-like gating can lead to better results. Why not adopt the gating structure in the proposed architecture? More discussions are required.

[1] TRANSFORMER BASED REINFORCEMENT LEARNING FOR GAMES.

[2] Stabilizing Transformer-Based Action Sequence Generation For Q-Learning.

Minor: Different structures in Section 5.3 can be illustrated in the supplementary material.

**Summary Of The Paper:**

The paper proposes a new transformer-based architecture to solve POMDP problems. The main contributions include using transformer decoder structure to address partially observable RL domains and utilizing intermediate Q-values for the training. Results on four environment sets are reported.

**Summary Of The Review:**

This paper has a good motivation and proposes a new solution for the POMDP problem. Experiments on four environment sets demonstrate its effectiveness. The major concern is the lack of comparison with state-of-the-art/similar approaches.

---

> ### Author Response · Authors · 2022-11-11
> **Response to reviewer oBw9**
>
> We thank the reviewer for their helpful comments. We hope to address the mentioned concerns below:
>
> > The proposed method is only compared with DQN and DRQN while there lacks evidence to show that DTQN can lead to state-of-the-art performance, which makes it less convincing.
>
> Our approach can be combined with other approaches such as more complex architectures or DQN extensions. However, such extensions would have made comparisons difficult; our goal with this work is to show the benefits of the transformer in DTQN. Our results show that DTQN never performs worse than DRQN and can often perform significantly better. We expect those results to similarly hold for extensions to our method, but those extensions are orthogonal to our contribution. We note that DRQN is the standard deep Q-based method for partially observable RL and is widely used.
>
> > More discussions and comparison results between [1] and DTQN should be provided. [2] also presents a very similar architecture (but it uses transformer encoder). More reviews and comparison results should be added.
>
> Both [1] and [2] use the transformer encoder whereas DTQN uses the transformer decoder. The main consequence of this design choice is those methods are unable to take advantage of intermediate Q-value prediction, since early timesteps can “cheat” and attend to future observations. Therefore, all other hyperparameters held the same, we expect those methods would display similar performance to DTQN without intermediate Q-value prediction from Table 1 and Figure 8 (appendix)
>
> > GRU-like gating can lead to better results. Why not adopt the gating structure in the proposed architecture? More discussions are required.
>
> In Table 1 and Figure 6 (Appendix), we notice the gating can lead to marginally better final performance. One possible explanation for this is the gating mechanism includes more trainable parameters than the residual skip connection, giving the network more power. For simplicity and connection to the original transformer, we decided to keep the residual skip connection as the “baseline” DTQN model, but we provide an open source and modular implementation (see updated supplementary) allowing researchers to change the “combine” step (as defined in Figure 1) with a simple command line argument. Regardless, all of these variants are potentially useful versions of our approach.

---

> ### Author Response · Authors · 2022-11-23
> **Response to reviewer oBw9**
>
> We hope to have addressed your concerns in our response, and encourage you to continue to join in the discussion so we may address any additional questions or concerns you may have. We have listed the major updates to our paper in the top-level comment [here](https://openreview.net/forum?id=cddqs4kvC20&noteId=SHB2K4TvbF). Thank you

---

### Official Review · Reviewer_zcAu · 2022-10-25

**Confidence:** 4
**Correctness:** 3
**Technical Novelty And Significance:** 2
**Empirical Novelty And Significance:** 2
**Recommendation:** 5

**Clarity, Quality, Novelty And Reproducibility:**

The paper is written clearly and results on synthetic POMDP tasks show improvement. I found the model to be derivative of previous work on transformers in RL. The results should be reproducible.

**Strength And Weaknesses:**

**Strengths** Applying transformers to POMDPs is an interesting direction. The empirical results improve upon previous baselines including LSTMs and DQN.

**Weaknesses**
1. The paper improves RNNs using Transformers in a straightforward way. It is not clear what is the main difference of the Transformer studied in this work compared to something like DT and variants. DT uses observations, actions, and rewards while DTQN uses only observations. Later variants improve DT using reward prediction, multi-task learning etc. I believe environments like Atari are also partially observable. Aside from some ablations, new insights that would be worth investigating further could really help.

2. Task that are studied are also synthetic. I think it is important to test the model on more challenging POMDPs even though current model fails to achieve strong performance.

3. Additional baselines that are tailored towards RL is also needed. At least DT, GATO, or variants.

4. Ablation experiments show that intermediate Q value prediction is important but it is not clear if this is due to simply using $k$ times more samples or actual intermediate Q value prediction. Are you using the same number of training samples for both experiments?


**Summary Of The Paper:**

This paper introduces deep transformer Q-networks (DTQN) for learning in partially observable environments. The authors train DTQN with a double Q-learning objective using on-policy samples. Given a trajectory of observations till now, observations are encoded using a transformer decoder with a masked attention to generate Q values for every step in trajectory. Typical Q learning loss with a fixed target network is applied to every step and the network is updated using the sum of losses from each step. When evaluated on several synthetic partially observable environments, DTQN is shown to perform better than LSTM variants including attention and traditional DQN. Ablation studies show that predicting Q values of intermediate steps is important for better performance, gating mechanism can help improve performance, and learning positional encodings is overall better but depends on the environment. Visualizing attention probabilities shows that bottleneck states are attended more.

**Summary Of The Review:**

While results on synthetic POMDPs are interesting, the model is derivative of previous work and no realistic task is studied.

---

> ### Author Response · Authors · 2022-11-11
> **Response to reviewer zcAu**
>
> We thank the reviewer for their helpful comments. We hope to address the mentioned concerns below:
>
> > It is not clear what is the main difference of the Transformer studied in this work compared to something like DT and variants. DT uses observations, actions, and rewards while DTQN uses only observations.
>
> Decision Transformer is trained using supervised learning whereas DTQN is trained with reinforcement learning. DTQN learns a Q-function whereas DT learns to clone the policy it was trained on.
>
> Architecturally, both DTQN and DT use the transformer decoder, with position encodings and causally masked self-attention. DTQN uses the agent’s observation history as input, whereas DT uses the agent’s observation and action history as well as the agent’s current Return-to-go. The return-to-go token is not suitable for our case (that is, online RL) because we do not know the agent’s future return while it is rolling out a trajectory. We will make these differences more clear in the paper.
>
>
> > I believe environments like Atari are also partially observable… Task that are studied are also synthetic. I think it is important to test the model on more challenging POMDPs even though current model fails to achieve strong performance.
>
> The domains in the paper *are* challenging POMDPs. This is clear from the fact that even DRQN fails to solve many of the tasks. We chose our domains to be representative of challenging partially observable tasks. While domains with images appear more complicated, their difficulty is often learning feature representations from images rather than learning to remember important information. The DQN team showed that stacking 4 frames of an Atari game is all that’s needed to make most games fully observable.
>
> > Additional baselines that are tailored towards RL is also needed. At least DT, GATO, or variants.
>
> Decision Transformer, GATO, and variants are trained via offline RL and therefore are not good baselines for DTQN, which is an online RL approach. Complex works like GTrXL [1] and COBERL [2], could apply but without an open-source implementation, lack reproducibility. We are currently running experiments on a variant of our methods that resembles GTrXL and we can include the results in a few days. Regardless, DRQN is the standard deep Q-based method for partially observable RL and we show that DTQN can significantly outperform it.
>
> [1] Parisotto, Emilio, et al. "Stabilizing transformers for reinforcement learning." International conference on machine learning. PMLR, 2020.
>
> [2] Banino, Andrea, et al. "Coberl: Contrastive bert for reinforcement learning." arXiv preprint arXiv:2107.05431 (2021).
>
> > Ablation experiments show that intermediate Q value prediction is important but it is not clear if this is due to simply using times more samples or actual intermediate Q value prediction. Are you using the same number of training samples for both experiments?
>
> In Table 1, we keep the batch sized fixed at 32 for both the variant of DTQN with and without intermediate Q-value prediction. To analyze performance with varying amounts of training samples, we ran a new experiment where the DTQN variant without intermediate Q-value prediction has a larger batch size (see Figure 9 of the updated supplementary materials). Despite increasing the agent’s batch size by a factor of 16, the performance during training is unstable without intermediate Q-value prediction. Training on q-values with little to no context produces a more stable and robust agent than only training on full contexts.

---

> > ### Comment · Reviewer_zcAu · 2022-11-28
> > **Thank you for the clarifications**
> >
> > I appreciate the authors' response. While the authors highlighted the difference of their work from others, the use of Transformers is still straightforward which is my main concern.
> >
> > Aside from applying transformers in partial observability context, which I believe is important but doesn't warrant a significant contribution in its current form, could you clarify what is the difference of your work from previous works that use a recurrent architecture in online RL such as R2D2?

---

> > > ### Author Response · Authors · 2022-12-09
> > > **Thank you for your question**
> > >
> > > Thank you for your question. We are glad we were able to clarify some earlier concerns.
> > >
> > > We use DTQN to solve POMDPs, introduce intermediate Q-value prediction to train on all generated q-values, and run several ablations to compare the different architectural choices within the transformer. Notably, we find intermediate q-value prediction to be the most important choice; we find that the lack of this training regime results in poor and unstable performance. We also visualize attention weights from our trained DTQN models and show that DTQN attends strongly to meaningful observation in its history, indicating the agent understands what it needs to know in order to solve the domain.

---

> ### Author Response · Authors · 2022-11-23
> **Response to reviewer zcAu**
>
> We hope to have addressed your concerns in our response, and encourage you to continue to join in the discussion so we may address any additional questions or concerns you may have. We have listed the major updates to our paper in the top-level comment [here](https://openreview.net/forum?id=cddqs4kvC20&noteId=SHB2K4TvbF). Thank you

---

### Official Review · Reviewer_CoHr · 2022-11-04

**Confidence:** 5
**Correctness:** 2
**Technical Novelty And Significance:** 1
**Empirical Novelty And Significance:** 1
**Recommendation:** 1

**Clarity, Quality, Novelty And Reproducibility:**

- quality: low
- clarity: low
- originality: low

**Strength And Weaknesses:**

Weaknesses:
1. Incremental work. Replacing RNN with Transformer in deep reinforcement learning is not a new idea! There are so many works with similar ideas, such as Decision Transformer[1], GATO[2], [3], [4], [5], [6] and [7].
2. Bad writing. The figures in this paper are really bad. For example, in Figure 2(a)(c), the blue curves are out of the figure.
3. Toy benchmarks. The authors claim that they are better than DQN or DRQN, they should conduct experiments on popular RL benchmarks, such as Atari, MuJoco, and DeepMind Lab.
4. Weak baselines. They should compare their method with real SOTA methods, such as PPO, Rainbow and IMPALA.


- [1] Chen L, Lu K, Rajeswaran A, et al. Decision transformer: Reinforcement learning via sequence modeling[J]. Advances in neural information processing systems, 2021, 34: 15084-15097.
- [2] Reed S, Zolna K, Parisotto E, et al. A generalist agent[J]. arXiv preprint arXiv:2205.06175, 2022.
- [3] Laskin M, Wang L, Oh J, et al. In-context Reinforcement Learning with Algorithm Distillation[J]. arXiv preprint arXiv:2210.14215, 2022.
- [4] Parisotto, Emilio, et al. "Stabilizing transformers for reinforcement learning." International conference on machine learning. PMLR, 2020.
- [5] Banino, Andrea, et al. "Coberl: Contrastive bert for reinforcement learning." arXiv preprint arXiv:2107.05431 (2021).
- [6] Furuta, Hiroki, Yutaka Matsuo, and Shixiang Shane Gu. "Generalized decision transformer for offline hindsight information matching." arXiv preprint arXiv:2111.10364 (2021).
- [7] Micheli, Vincent, Eloi Alonso, and François Fleuret. "Transformers are sample efficient world models." arXiv preprint arXiv:2209.00588 (2022).


**Summary Of The Paper:**

This paper proposed a transformer-based DQN agent.

**Summary Of The Review:**

This paper proposed a transformer-based DQN agent.

---

> ### Author Response · Authors · 2022-11-11
> **Response to reviewer CoHr**
>
> > Incremental work. Replacing RNN with Transformer in deep reinforcement learning is not a new idea! There are so many works with similar ideas, such as Decision Transformer[1], GATO[2], [3], [4], [5] and [6].
>
> This is incorrect. The work of [1], [2], [3], and [6] train transformers using supervised learning, not temporal difference learning, i.e not RL. Essentially, these methods are cloning a policy (a mapping from state to action) implicit in the data onto the transformer model. This is completely different from RL where we use a TD target to learn from trial and error.
>
> GTrXL [4] and CoBERL [5] could apply to our setting but since they do not release source code, we could not directly benchmark against those methods. Nevertheless, we did compare with a variant of DTQN that resembles GTrXL (Gate and Identity from Table 1), except that it incorporates our intermediate Q-value prediction method. Since DTQN performs at least as well as this variant, we expect DTQN to outperform the baseline GTrXL model (our inference time is also faster). We did not evaluate Transformer XL, but since any of the methods mentioned above could be modified with the Transformer XL model, we view this as an orthogonal choice. We will clarify these details in the final version of the paper.
>
> > Toy benchmarks. The authors claim that they are better than DQN or DRQN, they should conduct experiments on popular RL benchmarks, such as Atari, MuJoco, and DeepMind Lab.
>
> Again, we are focused on partially observable RL while those benchmarks are typically used for fully observable RL. In particular, mujoco is fully observable (and continuous control so a poor fit for Q-based methods) while stacking frames is sufficient in most Atari games. DeepMind Lab has some potentially partially observable tasks but it isn’t clear how partially observable they are. Our domains were designed specifically to be challenging partially observable tasks. While domains with images appear more complicated, their difficulty is often learning feature representations from images rather than learning to remember important information. Our goal was to use domains where the key difficulty is in the partial observability, not in other aspects that are not our research focus such as high-dimensional input.
>
> > Weak baselines. They should compare their method with real SOTA methods, such as PPO, Rainbow and IMPALA.
>
> If we were trying to solve MDPs, then comparing with PPO, Rainbow, etc. would make sense. However, we are not. Those methods are not appropriate to solving POMDPs, which is the problem we tackle in this paper. Those algorithms, like DQN, have no mechanism for remembering prior observations. Our approach (and our baselines) can be combined with other approaches such as more complex architectures or DQN extensions. However, such extensions would have made comparisons difficult; our goal with this work is to show the benefits of the transformer in DTQN. Our results show that DTQN never performs worse than DRQN and can often perform significantly better. We expect those results to similarly hold for extensions to our method, but those extensions are orthogonal to our contribution.
>
> [1] Chen L, Lu K, Rajeswaran A, et al. Decision transformer: Reinforcement learning via sequence modeling[J]. Advances in neural information processing systems, 2021, 34: 15084-15097.
>
> [2] Reed S, Zolna K, Parisotto E, et al. A generalist agent[J]. arXiv preprint arXiv:2205.06175, 2022.
>
> [3] Laskin M, Wang L, Oh J, et al. In-context Reinforcement Learning with Algorithm Distillation[J]. arXiv preprint arXiv:2210.14215, 2022.
>
> [4] Parisotto, Emilio, et al. "Stabilizing transformers for reinforcement learning." International conference on machine learning. PMLR, 2020.
>
> [5] Banino, Andrea, et al. "Coberl: Contrastive bert for reinforcement learning." arXiv preprint arXiv:2107.05431 (2021).
>
> [6] Furuta, Hiroki, Yutaka Matsuo, and Shixiang Shane Gu. "Generalized decision transformer for offline hindsight information matching." arXiv preprint arXiv:2111.10364 (2021).

---

### Comment · Reviewer_CoHr · 2022-11-12
**Comments on the rebuttals.**

The authors state that they do not compare their method with SOTA RL algorithms and popular benchmarks just because they want to solve POMDP tasks. This statement is not convincing! Reasons below:

(1) There are many Atari games that are POMDPs (e.g.,  Montezuma's Revenge), but the authors state that "stacking frames is sufficient". If the method is good enough, why did the authors just choose a toy grid world as the testbed? If you only evaluate your method on a simple self-built game, how can you verify the generality of the proposed method?

(2) The MDP is just a special case of POMDPs. If your method can work well on POMDPs, it will also work well on MDP.

(3) The authors say that they do not compare their method with PPO, because PPO can not solve POMDP. That's not correct! We can use RNN units in the neural network of PPO to capture historical information. Even if PPO can only work for MDP, it will be more convincing for the experimental results when the proposed method really outperforms previous SOTA methods.

I will keep my rating, and I suggest that other reviewers and ACs take a closer look at this paper. This paper has bad technical representations and insufficient experimental evaluations, which is definitively not good enough for a top conference.

---

> ### Author Response · Authors · 2022-11-14
> **Responding to comments from reviewer CoHr**
>
> The reviewers points are (again) incorrect. We give details below.
>
> (1) While some of the Atari games are partially observable by various degrees, these are not standard partially observable benchmarks. Therefore, we chose to use standard benchmarks and a new domain that are known to have difficult partial observability. As noted in our response, the Atari games have several other issues such as representation learning from images and exploration (e.g., Montezuma's Revenge) that would confuse results about partial observability.
>
> (2) MDPs are special cases of POMDPs but it doesn’t make sense to use standard MDPs as benchmarks. It is well known that by using the additional history, POMDP-based solutions will typically require more time to train and converge less robustly on standard MDPs. But comparing POMDPs and MDPs is not the focus of our paper.
>
> (3) Of course one could add recurrent layers to PPO, but one could add recurrent layers to any MDP algorithm to solve partially observable tasks. This is exactly what we did with DQN. Since our method is DQN-based, we compare with recurrent DQN (DRQN) as the closest recurrent method. If we wanted to compare with PPO, we could make a policy gradient version of our approach but this is outside the scope of our paper. We could include results for nonrecurrent PPO but we expect it to perform similarly to DQN, whose poor results show that the methods that don’t use history (e.g., PPO) would perform poorly on these partially observable tasks.

---

> > ### Comment · Reviewer_CoHr · 2022-11-27
> > **The authors' points are (definitively) incorrect. I give details below:**
> >
> > The authors' points are (definitively) incorrect. I give details below:
> >
> > (1) I have done some research work on POMDP, and I'm sure that Atari games can be used as POMDP testbeds.  You can have a look at these papers: [1][2][3]. If you want to show the effectiveness of your approach, please conduct experiments on Atari.
> >
> > (2) I'm not saying that you should compare POMDPs and MDPs. What I mean is that MDP is just a special case of POMDPs, if your method works well on POMDPs, it should work not badly on MDPs. This is very simple logic!
> >
> > (3) The authors say that making a PPO version of their approach is outside their scope. This is totally unconvincing! `Firstly`, this paper is not a theoretical paper, which means that their work should focus on empirical performance. As PPO is a classical and powerful DRL method, it's the author's duty to implement and evaluate such a method. `Secondly`, ICLR is a top machine learning conference, and the paper with empirical study should produce SOTA performances. DQN is an `old` value-based RL method, performance improving on such series is not convincing. The author should compare their method with more SOTA baselines to verify the effectiveness of their method.
> >
> >
> > - [1] Hausknecht, Matthew, and Peter Stone. "Deep recurrent q-learning for partially observable mdps." 2015 aaai fall symposium series. 2015.
> > - [2] Zhu, Pengfei, et al. "On improving deep reinforcement learning for pomdps." arXiv preprint arXiv:1704.07978 (2017).
> > - [3] Igl, Maximilian, et al. "Deep variational reinforcement learning for POMDPs." International Conference on Machine Learning. PMLR, 2018.

---

> > > ### Author Response · Authors · 2022-12-09
> > > **Responding to reviewer CoHr**
> > >
> > > We have responded to these points in previous comments in this thread, and still disagree with the reviewer. In summary:
> > >
> > > 1.) We chose our domains specifically to focus on partial observability and information gathering. While some Atari games may have some levels of partial observability, they include other issues which would confuse the results regarding the ability of our method to encode an agent's history.
> > >
> > > 2.) The focus of our paper is to use the transformer as a history encoder to solve partially observable domains. Experimenting with fully observable environments does not contribute to this goal.
> > >
> > > 3.) Because our domains require some form of memory in order to perform well, we expect a PPO version of DTQN to outperform classic PPO similarly to how DTQN outperforms classic DQN. This extension would be orthogonal to our contribution.

---

> > > > ### Comment · Reviewer_CoHr · 2022-12-14
> > > > **Responding to authors**
> > > >
> > > > The authors' response is groundless.
> > > >
> > > > 1. I have listed many papers, which study POMDPs and conduct experiments on Atari. If you really propose a good RL algorithm, please evaluate your method on popular benchmarks. Just giving nonsense reasons to avoid justified evaluation can not improve the quality of your paper!
> > > >
> > > > 2. Maybe you still don't understand the logic.  Your method is designed for POMDP, MDP is a subset of POMDP. You say that your method is generally good on POMDP, so I expect your method should not have a bad performance on MDP!
> > > >
> > > > 3. Your paper is an empirical study paper, all your statements should be supported by experimental results. And your transformer is designed for DQN, which is a value-based RL algorithm and also an off-policy algorithm; I don't think it is that easy to adapt to PPO, which is a policy-based algorithm and an on-policy algorithm. If you think it is trivial, please implement a PPO-version of your method!

---

### Author Response · Authors · 2022-11-16
**Updates to our paper**

We are thankful for the constructive feedback we've received. Since the beginning of the discussion phase, we have made the following updates:

- Added an experiment (Figure 9, Appendix) which aims to better understand the value of intermediate Q-value prediction by comparing our baseline DTQN agent against variants that have increased batch size but do not train with intermediate Q-value prediction. The results show that while increasing batch size generally improves performance, the baseline DTQN agent is still the best performing and is the only stable curve able to maintain a consistent ability to solve the domain. Our intermediate Q-value prediction challenges the agent to train on short sub-histories, including those which have little to no context, and produces a better and more robust agent.

- Added an experiment (Figure 10, Appendix) which compares the baseline DTQN agent with the “Gate and Identity” transformer structure from Table 1 but without intermediate Q-value prediction. This variant incorporates the gated combine step and identity map reordering proposed in GTrXL [1]. The baseline DTQN agent achieves better final performance than the gate and identity variant without intermediate Q-value prediction across all three domains. The difference is most notable in the Memory Cards domain, where this variant completely fails to solve the task.

- Source code is now provided in the supplementary materials

We hope these additional results will clarify some of the remaining questions and discussion points brought up in the reviews.

[1] Parisotto, Emilio, et al. "Stabilizing transformers for reinforcement learning." International conference on machine learning. PMLR, 2020.

---

### Decision · Program_Chairs · 2023-01-20

**Decision:**

Reject

**Justification For Why Not Higher Score:**

Important concerns remain regarding the empirical evaluation and placement in comparison with the state of the art.

**Justification For Why Not Lower Score:**

N/A

**Metareview: Summary, Strengths And Weaknesses:**

The paper addresses the problem of learning in partially observable reinforcement learning settings. The authors propose a deep transformer architecture that builds on deep Q networks. The approach is evaluated in several partially observable domains, including a standardized maze task and a novel memory cards domain.

Reviewers appreciated the importance of the problem addressed in this paper. The proposed approach is well motivated. Empirical results show improvements over LSTM and DQN-based baselines, and results suggest that the proposed intermediate Q-value prediction helps stabilize training.

At the same time, reviewers raise several concerns. The proposal of integrating transformers in an online RL approach is not novel - many variants have been proposed. Reviewers feel that the paper falls short in comparison to the latest state of the art approaches in this area, and that evaluation is performed on a set of domains with little to no overlap with state of the art approaches. As a result, the novelty and contribution of the work falls short. In addition, reviewers suggested that additional analysis could help generate deeper insights, e.g., in the role that intermediate Q-value prediction may play in stabilizing training.

Authors addressed some of the reviewer concerns in the rebuttal phase. In particular, they added two experiments to the appendix, one to better understand the role of intermediate Q-value prediction, and one to compare the proposed agent to a "gate and identity" transformer similar to an existing approach. Further, there was a detailed discussion between the authors and one of the reviewers about the choice of empirical testbed.

In its current state, important concerns remain. Establishing proper comparisons between existing approaches and the present work remains challenging, as test beds differ and the comparison to GTrXL remains a comparison to a variant of the approach. The paper could be substantially strengthened by focusing on the elements of the approach that are novel, and systematically establishing (within the main paper) how these elements contribute to improving performance over state of the art approaches. Overall, the paper is not recommended for acceptance in its present form.